# Dynamics of SARS-CoV-2 Antibody Responses up to 9 Months Post-Vaccination in Individuals with Previous SARS-CoV-2 Infection Receiving Inactivated Vaccines

**DOI:** 10.3390/v15040917

**Published:** 2023-04-04

**Authors:** Jing Wang, Lei Huang, Nan Guo, Ya-Ping Yao, Chao Zhang, Ruonan Xu, Yan-Mei Jiao, Ya-Qun Li, Yao-Ru Song, Fu-Sheng Wang, Xing Fan

**Affiliations:** 1The First Affiliated Hospital of USTC, Division of Life Sciences and Medicine, University of Science and Technology of China, Hefei 230001, China; 2Department of Infectious Diseases, The Fifth Medical Center of Chinese PLA General Hospital, National Clinical Research Center for Infectious Diseases, Beijing 100039, China; 3Chinese PLA Medical School, Beijing 100853, China; 4The Second School of Clinical Medicine, Southern Medical University, Guangzhou 510515, China; 5Senior Department of Infectious Diseases, Fifth Medical Center of Chinese PLA General Hospital, Beijing 100039, China

**Keywords:** SARS-CoV-2, Omicron variant, COVID-19 convalescent, neutralizing antibodies, longitudinal study, inactivated vaccine

## Abstract

Humoral immunity confers protection against COVID-19. The longevity of antibody responses after receiving an inactivated vaccine in individuals with previous SARS-CoV-2 infection is unclear. Plasma samples were collected from 58 individuals with previous SARS-CoV-2 infection and 25 healthy donors (HDs) who had been vaccinated with an inactivated vaccine. The neutralizing antibodies (NAbs) and S1 domain-specific antibodies against the SARS-CoV-2 wild-type and Omicron strains and nucleoside protein (NP)-specific antibodies were measured using a chemiluminescent immunoassay. Statistical analysis was performed using clinical variables and antibodies at different timepoints after SARS-CoV-2 vaccination. NAbs targeting the wild-type or Omicron strain were detected in individuals with previous SARS-CoV-2 infection at 12 months after infection (wild-type: 81%, geometric mean (GM): 20.3 AU/mL; Omicron: 44%, GM: 9.4 AU/mL), and vaccination provided further enhancement of these antibody levels (wild-type: 98%, GM: 53.3 AU/mL; Omicron: 75%, GM: 27.8 AU/mL, at 3 months after vaccination), which were significantly higher than those in HDs receiving a third dose of inactivated vaccine (wild-type: 85%, GM: 33.6 AU/mL; Omicron: 45%, GM: 11.5 AU/mL). The level of NAbs in individuals with previous infection plateaued 6 months after vaccination, but the NAb levels in HDs declined continuously. NAb levels in individuals with previous infection at 3 months post-vaccination were strongly correlated with those at 6 months post-vaccination, and weakly correlated with those before vaccination. NAb levels declined substantially in most individuals, and the rate of antibody decay was negatively correlated with the neutrophil-to-lymphocyte ratio in the blood at discharge. These results suggest that the inactivated vaccine induced robust and durable NAb responses in individuals with previous infection up to 9 months after vaccination.

## 1. Introduction

Coronavirus disease 2019 (COVID-19) is an acute respiratory infectious disease caused by severe acute respiratory syndrome coronavirus 2 (SARS-CoV-2) virus [1,2]. It has spread rapidly throughout the world, causing more than 673 million confirmed COVID-19 cases and 6 million deaths as of 10 February 2023, since its first outbreak in Wuhan, China, in December 2019 [3]. SARS-CoV-2 variants of concern (VOC), such as Alpha, Beta, Gamma, Delta, and Omicron, have emerged intermittently [4]. The Omicron variant is driving the current surge of cases in most countries and is currently the dominant strain globally because of its strong transmissibility [5,6]. Recently, studies have shown that neutralizing antibodies (NAbs) against Omicron induced by a previous infection or vaccination were 36- to 40-fold lower than those against the original wild-type strain [7,8,9]. Although NAbs offer little protection against infection with the Omicron variant, they appear to protect against hospitalization and severe disease [10,11,12].

Natural infection and vaccination produce high levels of humoral and cellular immune responses mediated by memory B and T cells, which can control the virus and greatly reduce morbidity and mortality rates [13,14]. Currently, most of the global population is no longer immunologically naive owing to prior infection and/or vaccination [15]. This mosaic of “hybrid immune situations” will influence the immune responses elicited by booster doses of vaccine or reinfection, especially in the context of emerging Omicron subvariants [16].

Evaluating the characteristics and durability of the humoral immune response to SARS-CoV-2 is essential for understanding breakthrough infections and immune protection in individuals with COVID-19. Recently, several studies have shown that high-affinity and high-efficiency NAbs are produced by receptor binding domain (RBD)-specific memory B cells. These NAbs support protective immunity in patients who have recovered from COVID-19 [17,18,19]. The levels of NAbs and RBD-specific memory B cells remain relatively stable between 6 and 9 months after infection. They have been shown to have some activity against VOC, including Omicron [20,21]. mRNA vaccination increases the NAbs level and the number of Spike-specific B cells to a greater extent in individuals with previous SARS-CoV-2 infection than in SARS-CoV-2-naive individuals, but does not enhance somatic hypermutation in B-cell receptors (BCR) in patients with COVID-19 [22]. Although Omicron evades a large fraction of NAbs, two doses of an mRNA vaccine induce Omicron-specific B-cell expansion, and prior infection history dramatically enhances Omicron-specific B-cell levels [23]. Six months after vaccination, the NAb level in convalescents is still higher than those in healthy donors (HDs), but the memory-B-cell level in patients with previous infection is comparable to that in HDs, because the memory B cells in individuals with previous infection decline faster than those in HDs [20]. Although the immunogenicity of inactivated vaccines is not as strong as that of mRNA vaccines, inactivated vaccine recipients are protected from severe disease and death [24,25]. The inactivated vaccine promotes SARS-CoV-2-specific B-cell expansion and anti-Omicron NAb production in individuals with previous infection similar to that in HDs in the short term [26]. To date, the short-term dynamics of the humoral immune response in individuals with previous SARS-CoV-2 infection receiving inactivated vaccines have been well studied, but the long-term dynamics of the antibody responses are unclear.

In this study, we longitudinally assessed the antibody responses against SARS-CoV-2 wild-type and Omicron strains in serial blood samples collected from individuals with previous SARS-CoV-2 infection and HDs after receiving inactivated vaccines for up to 270 days, and explored potential factors affecting the rate of NAbs decline.

## 2. Materials and Methods

### 2.1. Human Subjects

Fifty-eight individuals with previous SARS-CoV-2 infection (SARS-CoV-2 Recovered) were enrolled at the Central Theater of the Chinese PLA General Hospital from 6 April 2021 to 11 May 2022. All individuals had been infected with the wild-type strain and hospitalized between 4 February and 17 March 2020. Inclusion criteria were as follows: age 18 years or older, diagnosis according to the World Health Organization interim guidance, and positive for SARS-CoV-2 nucleic acid using a polymerase chain reaction test. The exclusion criteria were as follows: a history of anaphylactic response to vaccine components; acute febrile illness or symptoms associated with COVID-19 vaccination; human immunodeficiency virus, Hepatitis B virus, Hepatitis C virus, or influenza virus infection; and declining consent to participate in the study. Individuals with previous SARS-CoV-2 infection received two doses of inactivated vaccine 28 days apart, 1 year after infection. As a control group, we enrolled 25 HDs who had received two doses of inactivated vaccine, at the Fifth Medical Center of PLA General Hospital. HDs received a third dose of inactivated vaccine 180 days after the first dose, with a 28-day interval between the first and second doses of the vaccine. Peripheral blood samples were collected pre-vaccination (T0), 90.5 ± 10.7 days (T1), 180.6 ± 12.4 days (T2), and 270.5 ± 18.9 days (T3) after the first dose of vaccine in participants with previous infection, and at the same sampling timepoints after the third dose of vaccine in HDs (Figure 1A). Individuals were included in this analysis if they had provided at least one blood sample. Detailed clinical information and routine blood tests of all patients were collected using the hospital’s electronic medical records system.

### 2.2. Plasma Sample Collection

Whole blood was collected in ethylenediaminetetraacetic acid (EDTA) anticoagulation tubes (BD Biosciences, San Jose, CA, USA). Plasma was collected after centrifugation at 2000 rpm for 10 min, aliquoted, and stored at −80 °C.

### 2.3. Antibody Measurement

Concentrations of NAbs and immunoglobulin G (IgG) antibodies against nucleoside protein (NP) and Spike S1 domain of the wild-type and Omicron strains were measured in plasma using a chemiluminescent immunoassay (iFlash-2019-nCoV antibody kit; Shenzhen YHLO Biotech Co., Ltd., Shenzhen, China), as previously described [27]. The iFlash-2019-nCoV Neutralization Antibody assay is a one-step competitive immunoassay using a direct chemiluminescence immunoassay. The RBD of the SARS-CoV-2 wild-type or Omicron BA.1 strain was coated on magnetic beads. Acridinium ester-labeled angiotensin-converting enzyme 2 (ACE2) was designed to compete with SARS-CoV-2 NAbs for the RBD. NAb level were calculated using an iFlash3000 Chemiluminescence Immunoassay Analyzer (Shenzhen YHLO Biotech Co., Ltd.). The iFlash-2019-nCoV IgG assay, a paramagnetic particle chemiluminescent immunoassay for qualitatively determining the IgG antibody to SARS-CoV-2 proteins in human plasma, was used to measure IgG antibody against NP or S1 domain of the SARS-CoV-2 wild-type or Omicron BA.1 strain. The S1 domain or NP antigens were coated onto magnetic beads. Acridinium ester-labeled mouse anti-human IgG antibody was used to measure the SARS-CoV-2-specific IgG level in the plasma. Antibody activity was determined in arbitrary units (AU), and the cut-off was set as 10 AU/mL.

### 2.4. Statistical Analysis

GraphPad Prism 9.0 (GraphPad, San Diego, CA, USA) and SPSS 26.0 (IBM Corp., Armonk, NY, USA) software were used for statistical analysis. Continuous variables were expressed as medians and interquartile ranges (IQRs), and two-group comparisons were performed using Wilcoxon’s signed-rank test for paired groups and the Mann–Whitney U test for unpaired groups. Categorical variables were reported as counts (%) and compared using the χ^2^ test or Fisher’s exact test. The correlations between variables were evaluated using Spearman’s rank correlation test and visualized using R project and Corrplot packages. Two-sided *p* values < 0.05 were considered statistically significant.

### 2.5. Ethics

This study was approved by the Ethics Committee of the Fifth Medical Center, Chinese PLA General Hospital (2020-013-D). All participants provided written informed consent.

## 3. Results

### 3.1. Characteristics of Participants with Previous SARS-CoV-2 Infection and Healthy Donors

A total of 260 peripheral blood samples were obtained from participants in COVID-19 and healthy donor cohorts after vaccination (Figure 1A). The characteristics of the two cohorts are shown in Table 1 and Figure 1B,C. In participants with previous SARS-CoV-2 infection, the median age was 55.5 years (range 17–73 years), and 32 (55.2%) participants were male. Forty-three participants (74.1%, 2 mild cases, 41 moderate cases) had mild or moderate disease, and the remaining 15 (25.9%) participants had severe disease. Twelve (20.7%) participants had a comorbid condition: seven (12.1%) had hypertension, three (5.2%) had diabetes, and two (3.4%) had gout. The CoronaVac and BBIBP-CorV vaccines were administered to 16 (27.6%) and 42 (72.4%) participants, respectively. The median duration of hospitalization was 15 days and the median interval between symptom onset and vaccination was 388 days. In the HD cohort, all participants received the CoronaVac inactivated vaccine; the median age was 30.0 years (range, 26–50 years), and 13 (52.0%) participants were male. Remarkably, only 7 individuals in the HD cohort had a full longitudinal follow-up, 10 individuals provided only one peripheral blood sample, and 8 individuals provided two samples.

### 3.2. Dynamic Profiles of Anti-SARS-CoV-2 Antibodies in Participants with Previous SARS-CoV-2 Infection and Healthy Donors after Vaccination

To explore the kinetics of antibody responses in individuals with previous SARS-CoV-2 infection and HDs induced by the inactivated vaccine, we analyzed the levels of NAbs, Spike S1-domain-specific antibodies against the wild-type and Omicron BA.1 strains, and NP-specific IgG antibodies against the wild-type strain at different timepoints (Figure 2).

First, NAbs against the wild-type strain were measured at all available timepoints (Figure 2A). Approximately 81% of participants with previous SARS-CoV-2 infection had detectable NAbs before vaccination, consistent with a previous study [17], and the positivity rate increased to 98% and the geometric mean (GM) of NAbs were substantially increased to 53.3 AU/mL at T1 (20.3 AU/mL vs. 53.3 AU/mL, *p* < 0.0001). The NAbs were still detectable in 96% of participants at T2 and in 90% of participants at T3, and their NAb levels declined to 41.2 AU/mL and 32.7 AU/mL, respectively. In contrast, 85% of participants in the HD cohort were positive at T1 and then decreased to 18% at T3. The NAbs in participants with previous infection after two doses of vaccine were higher than those conferred by three doses of vaccine in HDs at all timepoints, and the differences were most striking at T3 (Figure 2B). There was a significant decline in the NAb level between T2 and T3 in the HDs (21.8 AU/mL vs. 6.7 AU/mL, *p* = 0.0020), which was not seen in participants with previous SARS-CoV-2 infection (41.2 AU/mL vs. 32.7 AU/mL, *p* = 0.4651), indicating a durable NAb response in individuals with previous SARS-CoV-2 infection after vaccination with inactivated vaccine.

To study cross-reactivity to the Omicron strain elicited by vaccination, we next measured NAbs against the Omicron BA.1 strain in the two cohorts (Figure 2C). Cross-variant neutralization was observed in 44% of participants with previous SARS-CoV-2 infection, 1 year after infection. Two doses of vaccine provided further enhancement of the antibody levels (27.8 AU/mL vs. 9.4 AU/mL, *p* < 0.0001), and the positivity rate rose to 75%. Omicron NAb levels were largely stable between T2 and T3, and 67% and 57% of participants were seropositive at T2 and T3, respectively (GM: 21.4 AU/mL and 18.0 AU/mL, respectively). In contrast, only 45% of HDs developed an Omicron neutralization function after the third dose of vaccine, and the positivity rate declined to 18% at T3. The Omicron Nab positivity rate was higher in participants with previous SARS-CoV-2 infection than that in HDs after vaccination, at T1 and T3 (Figure 2D).

As the majority of the NAbs displayed RBD-specific binding activity, we assessed the antibody levels targeting the Spike S1 subunit, which contains the RBD domain (Figure 2E–H). Vaccination promoted wild-type and Omicron strain S1 IgG antibody levels (wild-type: 133.1 AU/mL vs. 345.8 AU/mL, *p* < 0.0001; Omicron: 43.1 AU/mL vs. 138.9 AU/mL, *p* < 0.0001), and their dynamics in individuals with previous SARS-CoV-2 infection and HDs were similar over time, maintaining a 90-100% positivity rate before and after vaccination (Figure 2E,G). There was no significant difference in the levels of the wild-type (Figure 2F) or Omicron strain S1 IgG antibodies (Figure 2H) in individuals with previous SARS-CoV-2 infection and HDs.

Last, we investigated wild-type strain NP-specific IgG antibody levels (Figure 2I). The GM of the NP-specific IgG antibody was 19.2 AU/mL at 1 year after infection, and the seropositivity rate was 72%. In response to two doses of the inactivated vaccine, NP-specific IgG antibodies were measured in 96–100% of individuals with previous SARS-CoV-2 infection at three available timepoints. The level of NP-specific IgG antibodies at T2 was significantly lower compared with the previous timepoint (58.7 AU/mL vs. 87.4 AU/mL, *p* < 0.0001), and then was maintained at T3. In the HD cohort, 60% of participants developed a NP-specific IgG antibody response after the third dose of vaccine, the GM was 15.7 AU/mL, and the NP IgG antibody levels declined to 4.1 AU/mL at T3. Samples from individuals with previous SARS-CoV-2 infection had significantly greater NP-specific IgG antibodies at all timepoints compared with the HD group (Figure 2J). The dynamics of NP-specific IgG antibodies were consistent with those of NAbs against wild-type and Omicron strains.

Of note, due to the limitation that only seven HDs completed three follow-up visits, we investigated them separately to rule out the bias induced by the small size of longitudinal HD samples (Appendix A). Seven HDs showed a similar pattern in the dynamic of the antibodies with all HDs, and the antibody levels were no different in the two groups.

We also investigated the effects of the type of vaccine, age, sex, and disease severity on SARS-CoV-2 antibody production induced by the inactivated vaccine. The results showed that these variables had little effect on SARS-CoV-2 antibody levels (Appendix A). As HDs received the CoronaVac vaccine, and the individuals with previous SARS-CoV-2 infection received either the CoronaVac or BBIBP-CorV vaccine, the SARS-CoV-2 Recovered individuals were further grouped by the vaccine they received. The results showed that both subgroups of SARS-CoV-2 Recovered individuals had higher antibodies than HDs, and there was no significant difference in antibody levels induced by CoronaVac and BBIBP-CorV in individuals with previous SARS-CoV-2 infection (Appendix A). Antibody levels in the subgroups according to previous disease severity were similar at each timepoint, except for Omicron NAbs at T3 in the severe subgroup, which were slightly lower than those in the mild and moderate subgroups (Appendix A). Individuals aged >60 years old had slightly higher NP-specific antibodies against the wild-type strain at T0 than individuals aged ≤60 years (Appendix A).

### 3.3. Correlation Analysis between Antibody Levels in Individuals with Previous SARS-CoV-2 Infection and Healthy Donors after Vaccination

To assess how these different antibody responses induced by infection or vaccines interacted with each other over time, we created a correlation matrix to analyze the relationship among these five antibodies. We first determined the interconnections between different antibodies in 58 individuals with previous SARS-CoV-2 infection (Figure 3A). We observed strong mutual correlations between each antibody type before vaccination. This trend was also observed at other timepoints after vaccination, except for the NP-specific IgG antibody. NAbs targeting the wild-type or Omicron strain at T1 after vaccination were strongly associated with all the corresponding antibodies at T2 after vaccination, but weakly associated with those before vaccination. The correlation analysis in HDs showed a distinct pattern from that of participants with previous SARS-CoV-2 infection (Figure 3B). The Omicron S1 domain-specific IgG antibody was correlated at different timepoints in the HDs, but the correlation between the NAbs and S1 IgG antibodies against the wild-type strain at different timepoints was poor. No correlation was observed between NAbs against Omicron and NAbs against the wild-type strain in HDs.

### 3.4. Wild-Type NAb Duration after Vaccination Correlates with the Neutrophil-to-Lymphocyte Ratio

The levels of SARS-CoV-2 NAbs in individuals with previous SARS-CoV-2 infection and HDs declined over time. We quantified this decline by calculating the quotient of the NAbs against the wild-type strain at T3 divided by that at T1 for paired samples and defined this as the “NAbs durability index” [28]. Compared with individuals with previous SARS-CoV-2 infection, NAbs against the wild-type strain declined more rapidly after the third dose of vaccine in HDs (Figure 4A). The median NAbs durability index was 0.55 in participants with previous SARS-CoV-2 infection and 0.18 in HDs. These findings are consistent with a decaying neutralization response in most participants who received an inactivated vaccine [29].

To study the heterogeneity in NAb decline in individuals with previous SARS-CoV-2 infection after inactivated vaccine inoculation, participants with previous SARS-CoV-2 infection were grouped into “Sustainer” (NAbs durability index ≥ 1, *n* = 16, 27.6%) and “Decayer” (NAbs durability index < 1, *n* = 33, 56.9%) subgroups, based on the durability of their NAbs. Although the NAb levels against the wild-type strain in the Sustainers were slightly higher at T1 than at T0 (27.4 AU/mL vs. 15.5 AU/mL, *p* = 0.0215), they increased significantly at T3 (59.9 AU/mL vs. 27.4 AU/mL, *p* < 0.0001) (Figure 4B). In the Decayer group, the level of NAbs sharply increased from 26.8 AU/mL pre-vaccination to 73.3 AU/mL at T1, but they decreased to 24.0 AU/mL at T3. The level of NAbs in HDs declined from 33.6 AU/mL at T1 to 6.7 AU/mL at T3, similar to the Decayers. The level of NAbs in Sustainers was lower than that in Decayers at T1 (27.4 AU/mL vs. 73.3 AU/mL, *p* = 0.0011). This trend was also observed for other antibodies between the two groups at T0 and T1 (Figure 4C). However, 6 months later, the NAb level in the Sustainer group outnumbered the Decayer group at T3 (59.9 AU/mL vs. 24.0 AU/mL, *p* = 0.0040).

We next screened all available clinical parameters that may influence the NAbs durability index and found no significant relationship between age, hospital days, sex, and comorbid conditions (Table 2). To explore the influence between the previous severity of infection and the antibody response induced by vaccination, we drew a bar graph in which the Sustainers and Decayers presented in the mild and moderate and severe groups (Appendix A). The results showed that the composition of Sustainers and Decayers was the same in different disease severity groups. We found that the neutrophil-to-lymphocyte ratio (NLR) at discharge and 90 days post-vaccination (T1) was inversely correlated with the NAbs durability index, with r values of −0.3350 and −0.3477, respectively (Appendix A, Figure 4D). Lymphocyte counts and percentages were positively correlated with the NAbs durability index (Appendix A). NLR during the recovery stage may be used as a marker to predict the long-term NAb response of the inactivated vaccine in convalescents.

## 4. Discussion

Since the onset of the COVID-19 epidemic, the majority of the global population has acquired immunological memory owing to prior infection and vaccination [15]. Notably, long-lasting humoral immunity is central to viral clearance and protection from severe COVID-19 threats [30]. Although the inactivated SARS-CoV-2 vaccine is widely used worldwide, the long-term antibody responses in HDs and individuals with previous SARS-CoV-2 infection are unknown. In this study, we performed a longitudinal evaluation of antibody responses up to 9 months after inactivated SARS-CoV-2 vaccine inoculation in individuals with previous SARS-CoV-2 infection and HDs. The data suggest that the inactivated vaccine elicited robust, broad, and long-lasting NAbs in individuals with previous SARS-CoV-2 infection for up to 9 months. The NAb response at 3 months after vaccination was strongly correlated with the antibody response at 6 months after vaccination. The antibody levels declined rapidly in most individuals with previous SARS-CoV-2 infection who had a good NAb response after vaccination in a short time, and the NLR at discharge was identified as a marker for the NAb decay rate. The population needs to be closely monitored and administered an additional booster dose of vaccine.

In our study, all HDs received CoronaVac, and individuals with previous SARS-CoV-2 infection were vaccinated either with CoronaVac or BBIBP-CorV. Our data did not directly reflect the difference in antibody response induced by these two vaccines. A study suggested that compared to convalescents there was a low NAb response after the two-dose inoculation induced by BBIBP-CorV or CoronaVac, but there was no significant difference between the two vaccines [31]. Another study showed that CoronaVac vaccinees had higher antibody titers against Alpha than BBIBP-CorV vaccinees, whereas no significant differences in antibody titers against wild-type and other VOCs were observed between the two vaccines [32].

NAbs are essential for SARS-CoV-2 containment and the prevention of severe diseases. We found that compared with HDs, the NAb seropositivity rate against the wild-type and Omicron BA.1 strain was dramatically increased after inactivated vaccine inoculation, and these results are consistent with the antibody response induced by the mRNA vaccine in COVID-19 convalescents [20,23]. Although the NAbs decline gradually after infection, the level of Spike- or RBD-specific memory B cells increased in the first half year after natural infection and maintained a stable level within 1 year [33,34]. High-affinity NAbs are secreted by memory B cells or plasma cells, which develop in the germinal center (GC) in the lymph node after a cognate antigen encounter [35]. These SARS-CoV-2 memory B cells are capable of re-entering the GC, undergoing further affinity maturation, and mounting rapid and robust recall responses after re-exposure to vaccine or reinfection [23,36,37,38]. The affinity selection of memory B cells in GC, induced by infection or vaccination of mRNA or an inactivated vaccine, promotes the cross-reactive capacity of NAbs against new VOC variants, such as Omicron [39,40,41]. In this study, although these two cohorts received the wild-type SARS-CoV-2 virus-based inactivated vaccine, we found that participants with previous SARS-CoV-2 infection had higher and longer-lasting NAbs against Omicron than the HDs, which is consistent with the study with the mRNA vaccine [42]. A study of individuals with previous SARS-CoV-2 infection showed that prior SARS-CoV-2 virus infection enhances robust SARS-CoV-2 specific memory B cell development induced by mRNA vaccine inoculation [20]. COVID-19 convalescents receiving inactivated vaccines also have high levels of NAbs and SARS-CoV-2-specific memory B cells compared with those of HDs [26]. Notably, although individuals with previous SARS-CoV-2 infection and HDs were all exposed to the virus-specific antigen three times, their exposure interval differed. The Recovered cohort received two-dose booster vaccinations 1 year after the real infection, while the HD cohort received a booster vaccination 6 months after a simulated infection based on a two-dose initial vaccination. As was expected, different exposure intervals between the two cohorts led to different responses. However, the interval between the first dose and third dose of vaccine in HDs was shorter than that in the COVID-19 convalescents, and the antibody responses induced by vaccination in HDs were lower than those of Recovered individuals, so our conclusion is not affected.

Age, sex, and disease severity have been reported to impact the SARS-CoV-2 NAb titer in natural infections during the acute phase. In particular, NAbs and antigen-specific memory B cells are higher in severe SARS-CoV-2 infection and are positively associated with disease severity [43,44,45,46]. Our studies found that age, sex, and disease severity had little effect on antibody responses 1 year after infection. Another longitudinal study after infection showed that the magnitude of antibody response was higher in males, elders, or severe convalescents at the acute or earlier convalescent phase within 6 months; the difference was not observed at 1 year after symptom onset [47]. After vaccination, we found that the Omicron NAb level was lower in the severe disease subgroup than in the mild and moderate subgroups 9 months after vaccine inoculation. Although age is thought to be a prominent factor associated with vaccination efficiency, several studies have shown lower antibody levels in the elderly population [44]. In the context of COVID-19 convalescents, the magnitude, breadth, and affinity of SARS-CoV-2 antibodies induced by mRNA or inactivated vaccines are not influenced by age [48,49].

Most NAbs target the RBD domain of the Spike protein to block SARS-CoV-2 entry into the host cell, but NAbs against other Spike protein regions, such as the stem helix and fusion peptide regions in the S2 subunit, have also been identified [50]. Studies have shown that immunoglobulin A (IgA) dominates the early NAb response in SARS-CoV-2 infection and is replaced by IgG 28 days after symptom onset [51]. Thus, the NAb level is highly associated with IgG antibodies against the RBD or S1 domain in individuals with previous COVID-19 [30]. Our study found a correlation between NAbs and S1 IgG at each visit timepoint. In contrast, this relationship in HDs was entirely different. The level of NAbs targeting the wild-type or Omicron strain at 3 months after vaccination was strongly correlated with that at 6 months after vaccination, but poorly associated with that before vaccination. This result indicates that the NAbs in convalescents before or after vaccination may come from a different memory B cell population. Circulating memory B cells with expanded affinity-matured BCR in convalescents are preferably recalled by vaccination. They may enter secondary germinal centers to undergo further affinity maturation and contribute to peripheral NAbs [23,36]. In contrast, the NAbs in convalescents before vaccination were produced by memory B cells with a wide range of affinities [22]. BCR sequencing data also demonstrated that preferential V gene usage is prominently induced by natural infection or inactivated vaccines [52].

NAbs waning over time have been broadly reported in COVID-19 convalescents and vaccinated populations [53]. However, we found that some participants with previous SARS-CoV-2 infection maintained their NAbs against the wild-type strain after vaccination. NAbs and S1 IgG antibodies against wild-type or Omicron strains tended to be lower in the Sustainer group than those in the Decayer group at baseline and 90 days after vaccination, indicating that the frequency of memory B cells might be slightly inferior in the Sustainer group. Studies have shown that patients with severe COVID-19 have a higher level of memory B cells [33]. A study on the mRNA vaccine in COVID-19 convalescents shows that the level of Spike-specific memory B cells induced by the mRNA vaccine in HDs is lower than that in COVID-19 convalescents in the short term, but the Spike-memory B cell in convalescents declines quickly, and the memory-B-cell level in HDs even increases and is comparable to that of the COVID-19 convalescents at 6 months after vaccination [20]. Through the trend, we speculate that the memory-B-cell level in HDs may exceed that of individuals with previous SARS-CoV-2 infection 6 months after vaccination. We found that NLR at discharge and 90 days post-vaccination was negatively associated with the duration of NAbs. According to a previous study, NLR is an important prognostic indicator for COVID-19 patients, and patients with a higher neutrophil percentage and lower lymphocyte percentage are more likely to have a poor outcome [54]. Another study showed that the NLR was significantly higher in patients with a low level of S, RBD, and N-specific IgG than in patients with higher IgG levels [55]. Our supplemental data showed that Omicron NAbs were higher in the severe subgroup at T1 and declined to a lower level at T3 compared with the mild and moderate subgroups. This implies that severe disease may contribute to the higher fading rate of NAbs in the Decayer group. The mechanism of NAb duration needs to be investigated in the future.

Our study had several limitations. First, blood samples were collected up to 90 days rather than 7–28 days after receiving the first dose of vaccine; therefore, we could not perfectly explore the kinetics and characteristics of antibody responses shortly after vaccination. Second, the small size of the longitudinal sample, lack of samples before the third dose of vaccine, and mismatched age in HDs limit our conclusions. Finally, because of the availability of the sample source, we could not detect antibody levels in the lungs and air tracts. Our findings need to be further investigated in a well-designed cohort study with matched controls.

## Figures and Tables

**Figure 1 viruses-15-00917-f001:**
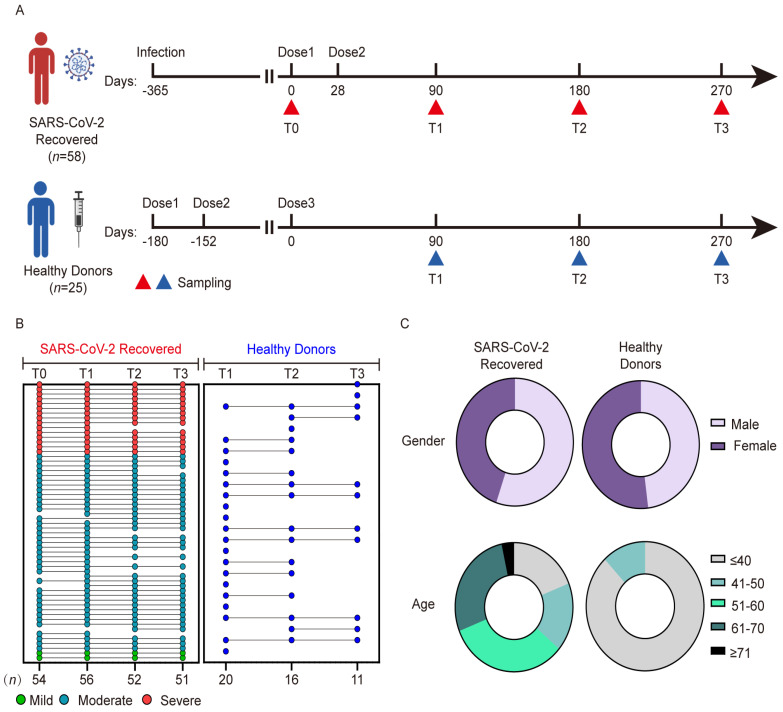
Individuals with previous SARS-CoV-2 infection and healthy donor cohorts and study design. (**A**) Study design of the vaccine cohort. Red and blue triangles indicate sampling collections, the numbers above the triangles indicate the days after the vaccination, and the timepoint for sampling is below the triangles. (**B**) Participants received the SARS-CoV-2 inactivated vaccine and donated blood at four visits. Disease severity status in individuals with previous SARS-CoV-2 infection is color-coded: mild (light green), moderate (dark green), or severe (red). Healthy donors are shown in blue. The bottom row displays the number of samples collected for each timepoint. (**C**) The pie plots show the distribution of sex and age in the two cohorts.

**Figure 2 viruses-15-00917-f002:**
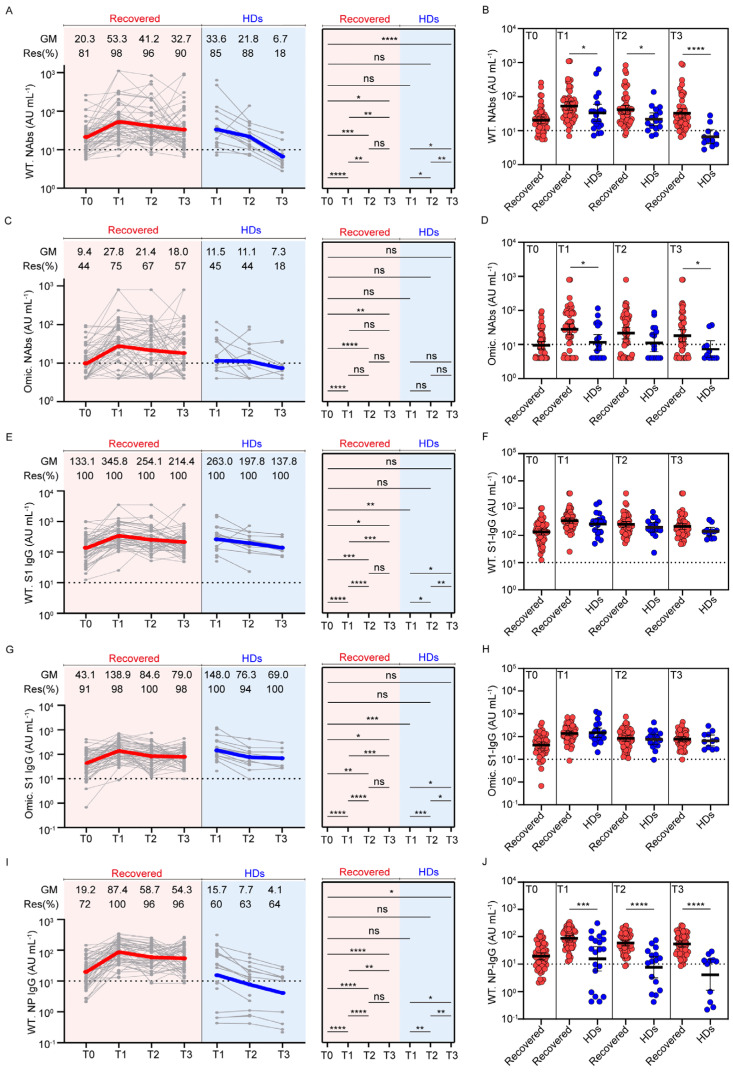
Dynamic changes of virus-specific antibodies in individuals with previous SARS-CoV-2 infection and healthy donors. The left parts show dynamic changes in NAbs against wild-type strain (**A**), NAbs against Omicron (**C**), Spike S1 domain-specific antibodies against wild-type strain (**E**), Spike S1 domain-specific antibodies against Omicron (**G**), and nucleoside protein-specific IgG antibodies against wild-type strain (**I**) in two cohorts over the different timepoints. Individuals are shown as gray symbols with connecting lines for longitudinal samples. Geometric means are shown in thick, colored lines: individuals with previous SARS-CoV-2 infection are color-coded in red and healthy donors in blue. The right parts show the comparison of antibodies between different time points. The scatter diagrams show the comparison of NAbs against wild-type strain (**B**), NAbs against Omicron (**D**), Spike S1 domain-specific antibodies against wild-type strain (**F**), Spike S1 domain-specific antibodies against Omicron (**H**), and nucleoside protein-specific IgG antibodies against wild-type strain (**J**) between individuals with previous SARS-CoV-2 infection and healthy donors at 3, 6, and 9 months post-vaccination. Dotted lines indicate the cut-off value. Statistics were calculated using Wilcoxon’s signed-rank test for paired groups and Mann–Whitney U test for unpaired groups: * *p* < 0.05; ** *p* < 0.01; *** *p* < 0.001; **** *p* < 0.0001. GM, geometric mean; Res, responders; NAbs, neutralizing antibodies; WT, wild-type strain; Omic, Omicron.

**Figure 3 viruses-15-00917-f003:**
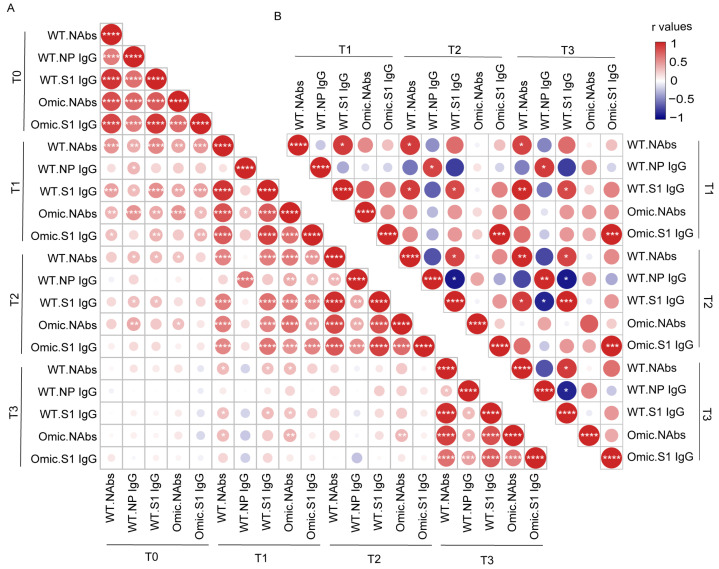
Correlation analysis between antibody levels in individuals with previous SARS-CoV-2 infection and healthy donors after vaccination. The correlation matrixes of antibody levels were calculated using nonparametric Spearman rank correlation in individuals with previous SARS-CoV-2 infection (**A**) and healthy donors (**B**). Positive correlations are shown in red and negative correlations are shown in blue. The size and color of each dot in the triangular matrix show the correlation strength between the variables: * *p* < 0.05; ** *p* < 0.01; *** *p* < 0.001; **** *p* < 0.0001. NAbs, neutralizing antibodies; WT, wild-type strain; Omic, Omicron.

**Figure 4 viruses-15-00917-f004:**
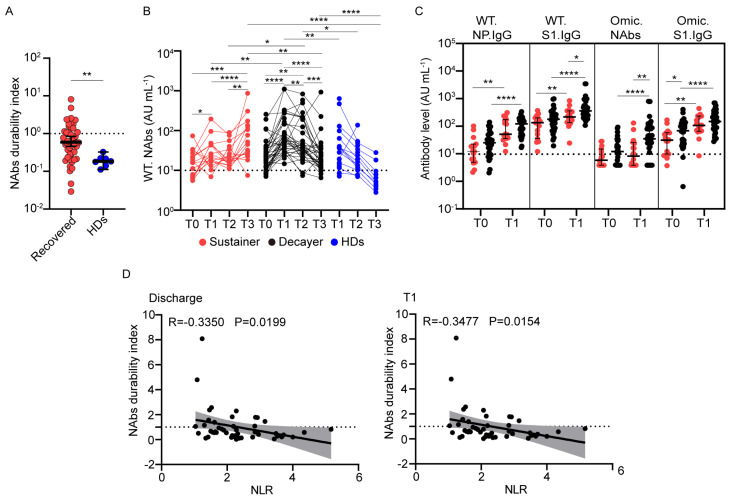
Association of clinical factors with wild-type neutralizing antibody duration after vaccination. (**A**) Dot plot shows the ranges of NAbs durability index for individuals with previous SARS-CoV-2 infection (red, *n* = 49) and healthy donors (blue, *n* = 7). (**B**) Wild-type NAb levels in the Sustainer group (red, *n* = 16), Decayer group (black, *n* = 33), and healthy donors (blue, *n* = 7) after vaccination. (**C**) Dot plot showing the antibody levels in the Sustainer group (red, *n* = 16) and Decayer group (black, *n* = 33) at 12 months after infection (T0) and 3 months post-vaccination (T1). (**D**) Spearman rank correlation was used to analyze the NAbs durability index and the neutrophil-to-lymphocyte ratio at discharge (left) and 3 months after vaccination (T1) (right). The linear fitting was performed, and gray indicates 95% confidence intervals. Statistics were calculated using Wilcoxon’s signed-rank test for paired groups and Mann–Whitney U test for unpaired groups: * *p* < 0.05; ** *p* < 0.01; *** *p* < 0.001; **** *p* < 0.0001. NAbs, neutralizing antibodies; WT, wild-type strain; Omic, Omicron; NLR, neutrophil-to-lymphocyte ratio.

**Table 1 viruses-15-00917-t001:** Characteristics of SARS-CoV-2 Recovered individuals and healthy donors in the study.

	Healthy Donors Cohort	SARS-CoV-2 Recovered Cohort
Total, *n*	25	58
Sex		
Female, *n* (%)	12 (48.0)	26 (44.8)
Male, *n* (%)	13 (52.0)	32 (55.2)
Age (years), Median (range)	30 (26, 50)	55.5 (17, 73)
≤60, *n* (%)	25 (100.0)	40 (69.0)
>61, *n* (%)	0 (0.0)	18 (50.0)
Race/Ethnicity		
Asian, *n* (%)	25 (100.0)	58 (100.0)
Disease severity		
Mild, *n* (%)		2 (3.4)
Moderate, *n* (%)		41 (70.7)
Severe, *n* (%)		15 (25.9)
Any comorbidity index		
Hypertension, *n* (%)	NA ^1^	7 (12.1)
Diabetes, *n* (%)	NA	3 (5.2)
Gout, *n* (%)	NA	2 (3.4)
Hospital days, Median (range)		15 (5, 36)
Vaccine type		
CoronaVac, *n* (%)	25 (100.0)	16 (27.6)
BBIBP-CorV, *n* (%)	0 (0.0)	42 (72.4)
Time between infection and vaccine, (days), Median (range)		388 (335, 437)

^1^ NA, not available.

**Table 2 viruses-15-00917-t002:** Effect analysis of age, hospital days, sex, and disease severity on NAbs durability index in SARS-CoV-2 Recovered individuals.

	Patients	Sustainer Group	Decayer Group	X^2^	*p*
**Age**					
≤60	34	13	21	1.574	0.2097
>60	15	3	12		
**Hospital days**					
≤14	19	7	12	0.2476	0.6187
>14	30	9	21		
**Sex**					
Male	28	10	18	0.2784	0.5977
Female	21	6	15		
**Disease severity**					
Mild and Moderate	37	12	25	0.003344	0.9539
Severe	12	4	8		

## Data Availability

All data generated by the study are included in the manuscript or in its Appendix A.

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
