# Peer review of "Dynamics of SARS-CoV-2 Antibody Responses up to 9 Months Post-Vaccination in Individuals with Previous SARS-CoV-2 Infection Receiving Inactivated Vaccines"

_viruses, 2023, doi:10.3390/v15040917_

Round 1

Reviewer 1 Report

Please find attached my review as word document!

Reviewer 2 Report

The manuscript titled: “Dynamics of SARS-CoV-2 antibody responses up to 9 months post-vaccination in individuals with previous SARS-CoV-2 infection receiving inactivated vaccine.” submitted to the journal Viruses with manuscript ID: viruses-2252277 by Fu-Sheng Wang et al.  analyzed the antibody responses up to 9 months after receiving the inactivated SARS-CoV-2 vaccine of 58 people with previous SARS-CoV-2 infection and 25 healthy donors.

The paper fits the journal’s topic, has the potential to contribute to the knowledge about immune response induced by inactivated vaccines and is very well written.

The major weakness of the study is the design- the small sample size and mismatched control group. However, I found the data of individuals with previous SARS-CoV-2 infection valuable, giving significant information for antibody levels after vaccination with the inactivated vaccine for up to 9 months.

Some minor issues should be taken into account before publication, as follows:

Page 2, line 85: “after vaccination” should be replaced by “receiving inactivated vaccines” or similar

Page 3, line 142: Did the provision of verbal consent via telephone approved by the Ethics Committee? 

Page 4, line 152: Except for the age mismatching of the control group, the discordance of the type of vaccine should also be taken into account. All HDs received a Sinovac inactivated vaccine, while just 16 (27.6%) of the SARS-CoV-2 recovered cohort received the same vaccine. Since some studies have reported differences in the effectiveness of Sinovac and Sinopharm inactivated vaccines, this issue should be discussed and pointed out as an additional limitation.

Page 4, line 152: The detail of the received Sinovac and Sinopharm inactivated vaccines should be added. What type are they -  BBIBP-CorV; NVSI-06-07; Trivalent CoronaVac vaccines?

Page 10, Figure 4 and Table 2: The presented data are of 56 individuals with previous SARS-CoV-2 infection. Please, add the rest two or explain the reason for their missing.

Round 2

Reviewer 1 Report

The Authors addressed the many critic point raised in my initial peer review very thoughtfully and improved the manuscript to a quality that readers are enabled to make up their opinion about the data themselves. I am satisfied with the revision and can recommend this article for publication in Vaccines.